# Health Benefits of Consuming Foods with Bacterial Probiotics, Postbiotics, and Their Metabolites: A Review

**DOI:** 10.3390/molecules28031230

**Published:** 2023-01-27

**Authors:** Victor E. Vera-Santander, Ricardo H. Hernández-Figueroa, María T. Jiménez-Munguía, Emma Mani-López, Aurelio López-Malo

**Affiliations:** Department of Chemical, Food and Environmental Engineering, Universidad de las Américas Puebla, Puebla 72810, Mexico

**Keywords:** food, bacterial probiotics, postbiotics, lactic acid bacteria, health benefits

## Abstract

Over the years, probiotics have been extensively studied within the medical, pharmaceutical, and food fields, as it has been revealed that these microorganisms can provide health benefits from their consumption. Bacterial probiotics comprise species derived from lactic acid bacteria (LAB) (genus *Lactobacillus*, *Leuconostoc*, and *Streptococcus),* the genus *Bifidobacterium*, and strains of *Bacillus* and *Escherichia coli*, among others. The consumption of probiotic products is increasing due to the current situation derived from the pandemic caused by COVID-19. Foods with bacterial probiotics and postbiotics are premised on being healthier than those not incorporated with them. This review aims to present a bibliographic compilation related to the incorporation of bacterial probiotics in food and to demonstrate through in vitro and in vivo studies or clinical trials the health benefits obtained with their metabolites and the consumption of foods with bacterial probiotics/postbiotics. The health benefits that have been reported include effects on the digestive tract, metabolism, antioxidant, anti-inflammatory, anticancer, and psychobiotic properties, among others. Therefore, developing food products with bacterial probiotics and postbiotics is a great opportunity for research in food science, medicine, and nutrition, as well as in the food industry.

## 1. Introduction

Probiotics are defined as microorganisms that, when consumed in adequate doses or concentrations, can benefit the consumer’s health [1]. Zucko et al. [2] noted that the benefits of probiotics range from the relief of gastrointestinal disorders to help the treatment of allergies, obesity, depression, bacterial vaginosis, and the improvement of the gastrointestinal tract, among many others. Currently, a wide variety of microorganisms are considered bacterial probiotics, of which a considerable portion belongs to the group of lactic acid bacteria (LAB), mainly of the genus *Lactobacillus*, *Leuconostoc*, and *Streptococcus* [3,4]. In addition, other non-LAB microorganisms are also considered bacterial probiotics, including some species of bifidobacteria, *Bacillus*, and *Escherichia coli*, among others [4]. Figure 1 shows selected species that could be considered probiotics; however, the strain level should also be checked for specific health benefits.

Due to the recent COVID-19 pandemic, there has been concern about improving health wellbeing and avoiding the spread of the virus. Hence, society began to consume foods that could raise the immune system. Since foods with probiotics can contribute to health (including the immune system), there has been an increase in demand for these products. As a result, the global probiotics market in 2022 was 68.56 billion USD. Still, this value is estimated to increase to 133.92 billion USD by 2030, due to the high consumption of probiotic products triggered by the pandemic [5].

**Figure 1 molecules-28-01230-f001:**
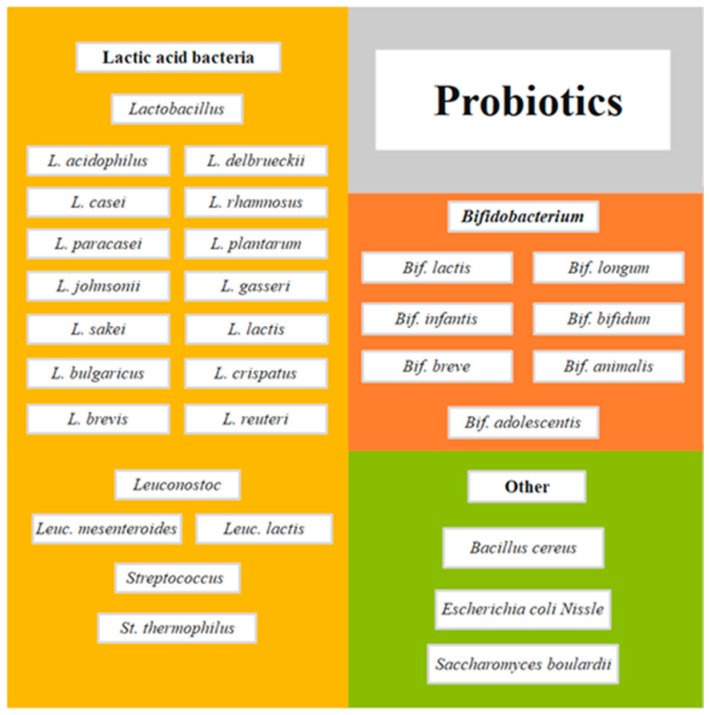
Selected species of microorganisms as probiotics adapted from Moradi et al. [6].

Furthermore, other factors such as increasing trends in consuming functional foods, new marketing strategies by social networks, and innovative food distribution channels such as online shopping can increase the demand for these products. Therefore, the food and pharmaceutical industries need research to offer humankind better products.

Probiotics are present in different forms for intake, either in drugs such as capsules, tablets, oral suspensions, and powders, or through nourishment. To obtain the benefits of probiotics, it is essential to consider the availability and accessibility of its products. Foods with probiotics are excellent means of offering them, as they can be coupled with the daily diet of society. In manufacturing food products, these can be intentionally incorporated with bacterial probiotics to improve their bioactivity, helping the treatment of specific pathologies (like functional foods) or consumer wellness. Foods related to probiotics include fermented products with probiotics, foods with encapsulated probiotics, and foods with bacterial postbiotics (inactivated bacteria with their metabolites derived from fermentation). Research is constantly advancing to understand and verify the beneficial effects of probiotics on the human body. The benefits that can be provided to the metabolites produced by bacterial probiotic fermentation and foods related to bacterial probiotics have been investigated. Therefore, this work presents a bibliographic compilation of the health benefits of food products incorporated with bacterial probiotics, being either fermented, microencapsulated, or inactivated, as well as the metabolites that are synthesized during fermentation. Studies in vitro, in vivo, and clinical trials show their consumption effects.

## 2. Food and Probiotic Bacteria

There is a great diversity of foods related to probiotics; for this reason, in this study, these foods are classified into three groups based on what is reported in the literature: (1) fermented foods with probiotics, (2) foods with encapsulated probiotics, and (3) postbiotic foods (new trend). Although this classification is presented in Figure 2, these food groups are detailed further below.

### 2.1. Fermented Foods

Fermented foods have been mainly related to probiotic bacteria. Therefore, these fermented products greatly interest the industry and scientific communities. This is because they comprise specific desirable properties, such as an extended shelf-life (owing to the acidification and reduction of the pH), an improvement of organoleptic properties (texture and production of aromas), increased nutritional value, and provide benefits to the consumer [7]. According to Ilango and Antony [8], fermented foods and beverages can be classified in different ways, either (1) by the microorganisms that ferment, (2) by the type of food or beverage, (3) the production method, (4) geographical localization or the producing and consuming communities of these foods. In the first type of fermented foods and beverages, there is a wide variety of fermentative species, of which the most used are the LAB. For example, in yogurt elaboration, lactic cultures include *Streptococcus thermophilus* and *Lactobacillus delbrueckii* ssp. *bulgaricus.* Another fermented food consumed worldwide is cheese, which, in some cases, is added with *Lactococcus lactis* and *Leuconostoc mesenteroides.* Other non-LAB bacteria have also been used to produce fermented legumes (*Bacillus subtilis* ver. *natto)* and cheese maturation (*Propionibacterium freudenreichii*) [9]. The second type of foods and beverages can be classified into dairy (yogurt, cheeses, sour cream, buttermilk, kefir, etc.), meat (salami, pepperoni, Serrano ham, etc.), vegetables and tubers (pickles, kimchi, sauerkraut, etc.), fruits (alcoholic beverages such as wine, fruit liqueurs, apple cider, some kinds of vinegar such as apples, grapes, etc.), cereals (beer, bread, and sake), legumes (natto, tempeh, miso, etc.), and in a mixture of foods, such as kimchi, that is sometimes fermented with vegetables and fish [9,10,11].

The third category of fermented foods and beverages based on the production method consists of modifying the food’s intrinsic and extrinsic factors to obtain the best growth conditions for the microorganisms. Among the inherent factors of the physicochemical composition of the food are the regulation of pH, salt concentration, and the addition of sugars and/or additives. In comparison, extrinsic factors modify environmental conditions, such as temperature and atmospheric composition. Fidan et al. [10] noted that intrinsic and extrinsic conditions are vital to obtaining a desired fermented product. For example, if a LAB is required to have antimicrobial properties, it must remain alive at 4 °C for a set amount of time [10]. Oh et al. [12] studied several growth conditions for their optimization, including temperature (15–55 °C), pH (2.5–8.0), and salt concentration (0–10% *w*/*v*) for *Lactobacillus plantarum* PMO 08. Their results showed that the optimal growth temperature was 30 °C, and conditions such as pH and salt concentration could be from 3.5–8.0 for pH and 0–8.0% NaCl. Finally, the fourth type of fermented foods and beverages was assigned by geographical location, or the producing and consuming communities of these foods is mainly based on the culture of each region depending on their traditions and customs, which influence their way of feeding.

In the present work, foods fermented with bacterial probiotics that include microorganisms of the LAB group and bacteria not belonging to this group, such as the genus *Bifidobacterium* and *Bacillus,* are reviewed. It is important to delimit the food models that are related to probiotics. Some fermented foods can benefit the consumer, but these foods or drinks do not contain probiotics. A microorganism must meet specific requirements to be considered a probiotic. Pineiro and Stanton [13] outlined several requirements. First, the microorganism with probiotic potential must be named according to the basis on which it can be understood anywhere in the world, such as the International Code of Nomenclature. In addition, it has been recommended that identification be carried out by official methods, such as PFGE DNA/DNA hybridization and 16S RNA sequencing. Subsequently, the microorganism must confer health benefits, which must be determined by in vivo, in vitro, and human tests. Finally, the probiotic must be analyzed for use in industry, evaluating shelf-life conditions and manufacturing practices, including probiotics clearly and concisely in the food labeling expressing the minimum dose contained and the health claims. In terms of regulation, a regulatory framework must be established, which includes strain, efficiency, security, labeling, possible fraud, and appropriate claims. If it is a product fermented with a probiotic, the specific health benefits of the product must be established [13]. Hence, those fermented foods or beverages that do not meet these requirements cannot be considered probiotic products, but they can be considered products with potential probiotic effects. Based on this idea, the term next-generation probiotics (NGPs) has emerged, which are candidate microorganisms to be new probiotics, since they have shown beneficial effects on human health. NGPs are species of the genus *Clostridium, Bacteroides, Faecalibacterium, Parabacteroides,* and *Eggerthellaceae* [14].

### 2.2. Microencapsulated Probiotic Food

Generally, foods containing microencapsulated probiotics do not need fermentation to provide their benefits, as the incorporation of probiotics is commonly carried out in non-dairy food matrices or those foods that conventionally do not provide benefits derived from probiotics [15]. Frakolaki et al. [13] determined that foods with encapsulated probiotics are known as “carrier probiotics” since they mainly transport the probiotics to the desired site, such as the gastrointestinal tract. However, the microencapsulation of probiotics is challenging, since their viability during ingesting must be guaranteed. This implies that the probiotic strains must remain stable and metabolically active during the product’s shelf-life. Furthermore, the encapsulation and the probiotic must be sufficiently resistant to the conditions of the gastrointestinal passage. The stomach is the most critical point because it is the most acidic organ of the human body, with a pH ranging from 1 to 2.5 [16,17].

For the microencapsulation of probiotics, it is essential to consider the food matrix, the probiotic strain, the encapsulation technique, and the encapsulating material [16]. In the food matrix, there are studies where encapsulated probiotics in foods, such as dairy [18,19,20,21,22], meat [23,24], non-dairy beverages [25,26,27], desserts [28,29,30], and bakery products [31,32,33,34] have been investigated. Tripathi and Giri [35] noted that lactobacilli are more resistant to various processes and storage conditions than bifidobacteria, which should be considered when choosing a probiotic strain. Lactobacilli show more resistance to low pH and a better adaptation to milk and other foods. For instance, *Lactobacillus casei* and *Lactobacillus acidophilus* have been shown to have resistance in gastric environments (simulation of gastric juice at pH 3.0 at a temperature of 37 °C), unlike *L. delbruekii* ssp. *bulgaricus,* which is not resistant [36]. Encapsulation techniques include extrusion, emulsification, coacervation, spray drying, freeze drying, fluidized bed drying, electrospinning, electro spraying, hybridization, and impact aerosol technology [16]. When choosing from these techniques, several factors must be considered, including the economic profitability of the technique (considering the available budget and costs of the equipment), the quality of the final product, and the operating conditions (low temperatures, suitable pH, low oxygen levels, and sterile environment) [37,38].

For the encapsulating agents, a wide variety of materials or biopolymers have been used, such as alginate, chitosan, inulin, starch, maltodextrins, pectins, carboxymethylcellulose, arabic gum, carob gum, xanthan gum, carrageenan, agar, polyols (sorbitol, mannitol, and xylitol), fatty acids, gelatin, zein, gluten, soy protein isolate, milk proteins, β-lactoglobulin, casein, whey, and cellulose acetate phthalate (CAP), among others [16,39,40]. The choice of encapsulating material involves various considerations, since it must be safe for the consumer (recognized as GRAS), resistant to the operating conditions and digestion of the human organism, and subsequently release microorganisms into the intestinal tract. It is proposed to encapsulate the probiotics with these coatings to raise the bioactivity of food. For example, Eratte et al. [41] studied a complex co-encapsulation of probiotic coacervates of tuna oil, whey protein isolates, and arabic gum. The encapsulated probiotic was *L. casei 431.* Their results demonstrated the addition of tuna oil in the dehydrated encapsulates improved the viability of the probiotic more than in the dry encapsulates without oil.

### 2.3. Postbiotic Food

The term postbiotic emerged from various investigations suggesting that probiotics’ viability is not necessary to confer health benefits [42,43,44,45]. Multiple researchers have provided several definitions or terminologies to describe postbiotics. Tsilingiri and Rescigno [46] defined postbiotics as “any factor ending the metabolic activity of a probiotic or any released molecule capable of conferring beneficial effects on the host directly or indirectly”. They have also been defined as compounds produced by microorganisms, released from food components or microbial constituents (including non-viable cells) that promote health and wellbeing when administered in adequate amounts [47]. Based on these definitions and that of other authors, such as Aguilar-Toalá et al. [42], Johnson et al. [48], and Wegh et al. [49], who proposed their own definitions, three important features regarding postbiotics can be determined. The first is that they are chemical compounds released or excreted by microorganisms, the second is that they have a bioactive effect on the host and microbiota, and the third is that the microorganisms that produced these substances are non-viable or have been inactivated. Today, there is no official definition of postbiotics, since it is still being studied which compounds and parts of non-living probiotics may belong to this definition. The most current terminology was given by the consensus panel of the International Scientific Association of Probiotics and Prebiotics (ISAPP), which defined the postbiotic as a “preparation of inanimate microorganisms and/or their components that confer a benefit to the health of the host” [50]. However, Aguilar-Toalá et al. [51] disagreed with a reply in which they argued that this concept is confusing due to the word “inanimate”, since this term is generally not used in the field of microbiology. They suggested changing it to “non-viable or inactivated” due to its extensive use in the scientific community.

Postbiotics have technological advantages over probiotics incorporated into foods, since they are more stable, and their storage, handling, and transportation can be more economically feasible [52]. In addition, Homayouni-Rad et al. [53] noted that postbiotics have better absorption through metabolism and a high signaling capacity to various organs and tissues, and can trigger several biological responses that benefit the consumer.

For the formulation of foods with postbiotics, the bacterial probiotic should be considered; the growth medium, either in the feed or a selective culture medium (such as MRS broth); growth conditions such as temperature, pH, substrates, and water activity, among others. In addition, the inactivation technique should be selected, either through conventional (pasteurization or sterilization) or emergent technologies (ultrasound, UV, high pressures, or irradiation), as well as the separation process consisting of centrifugation and filtration that may be optional. This is in addition to the concentration method, such as spray drying or freeze drying, and the incorporation of postbiotics into the food matrix, either as liquid or solid [6].

## 3. Health Effects of Metabolites from Probiotic Bacteria

The growth of bacterial probiotics is involved in fermentation, which metabolizes the nutrients of the medium and produces a wide variety of metabolic compounds. Metabolites from bacterial probiotics have been studied as they may be the main cause of positive effects on consumer health [42]. As mentioned above, most bacterial probiotics belong to the LAB group, which primarily synthesizes lactic acid from carbohydrate fermentation [54]. Bacterial probiotics excrete secondary metabolites such as organic acids, short-chain fatty acids, enzymes, peptides, teichoic acids, peptidoglycans, exopolysaccharides, vitamins, plasmalogens, neurotransmitters, biosurfactants, amino acids, and flavonoid-derived compounds such as desaminotyrosine, equol daidzein, noratirriol, terpenoids, and phenolic compounds, among others [45,55].

Metabolites from bacterial probiotics have been studied as they may be the main cause of positive effects on consumer health [42]. Figure 3 shows a diagram of the health effects that have been studied concerning bacterial probiotic metabolites. Various components have been investigated, including short-chain fatty acids (SCFAs), bacteriocins, enzymes, teichoic acids, exopolysaccharides (EPS), vitamins, plasmalogens, and biosurfactants.

### 3.1. Short-Chain Fatty Acids

SCFAs are the main metabolites produced by the fermentation of intestinal bacteria; these compounds are produced from the fermentation of non-digestible carbohydrates (prebiotics) by the intestinal microbiota [44]. The SCFAs primarily produced are acetate, propionate, and butyrate. The metabolic pathways of acetate are very diverse and depend on the type of bacteria, while the synthesis of propionate and butyrate depends on a specific substrate [56]. Gill et al. [57] determined that homo-fermentative bacteria synthesize SCFAs from carbohydrate fermentation to produce pyruvate via glycolytic, while heterofermentative bacteria, such as bifidobacterial, do so through the phosphoketolase pathway.

The bacterial probiotics that have been studied are mainly of the genus *Lactobacillus* and *Bifidobacterium*. For example, one study investigated the production of SCFA from *Bif. longum* SP 07/3, *Bif. bifidum* MF 20/5, *L. gasseri* PA, 16/8, and *L. rhamnosus* GG [58]. In the results, the highest production of acetate and propionate was with *Bif. Bifidum*, with a concentration of approximately 125 mM and 130 μM, respectively, while *L. rhamnosus* GG presented the lowest production of SCFA, with 55 mM of acetate and 100 μM of propionate. SCFAs activate G-specific protein-coupled receptors (GPRs), where GPR41 (known as free fatty acid receptor 3) and GPR43 (free fatty acid receptor 2) have been related to a wide health effect of antitumor and anti-inflammatory benefits on the colon, protection against the development of immune disorders, obesity control, glucose homeostasis control, appetite regulation, and cardiovascular effects [45,57,59]. Gill et al. [57] suggested the consumption of SCFA through the diet as a postbiotic effect, since it has been reported that vinegar and kombucha are the foods with the highest levels of SCFA, with 1015.82 mg of acetate per tablespoon (15 mL) and 1222.95 mg of acetate in a portion of 330 mL, respectively.

In comparison, some cheeses, such as Swiss cheese and blue vein cheese, are high in propionate (109.14 mg) and butyrate (136.54 mg) in a serving of 25 g, respectively [60]. A minimum dose of consumption of SCFAs has not been determined for functional health effects. However, an intervention study tested a beverage with apple cider vinegar containing 25.8 mM of acetate, 0.05 mM of propionate, and 0.04 mM of butyrate. The beverage helped reduce blood pressure, showing potential cardiovascular effects [60]. However, the doses of these compounds and how they can be supplied through food remain to be investigated more thoroughly. 

### 3.2. Plasmalogens

According to Řezanka et al. [61], plasmalogens are lipid compounds with important functions in organisms such as bacteria, protozoa, invertebrates, and mammals. The largest classes of plasmalogens are glycerophosphoryl ethanolamine and glycerophosphorylcholine with a vinyl ether in the position of glycerol *sn-1* instead of a fatty acid. At the position of glycerol *sn-2*, plasmalogens are enriched with a polyunsaturated fatty acid [62]. Regarding bacterial probiotics, only the production of these compounds with *Bif. animalis* subsp. *lactis* has been investigated. This research was conducted by Oberg et al. [63], in which plasmalogens were recognized as endogenous antioxidants induced by hydrogen peroxide (H_2_O_2_). The effects caused by these compounds are positively involved in neuro-regeneration and the alleviation of heart disease; it has positive effects against type 2 diabetes, obesity, inflammation, and cancer [62].

Regarding neuro-regeneration, plasmalogens are the main components of neuronal tissue, contributing between 50% and 80% of the total glycerophosphoryl ethanolamines of the gray and white matter of the brain [64]. As proof of this claim, Hossain et al. [65] observed that a daily intake of plasmalogens could improve cognition in Alzheimer’s patients. However, research into these metabolites is still very early, and more research is needed concerning their production in bacterial probiotics and their potential health benefits.

### 3.3. Enzymes

Enzymes excreted by microorganisms have a variety of biochemical, physiological, and regulatory functions. As shown in Figure 3, the health benefits of proteases are the protection of cells from oxidative stress and effects against heart disease and cancer [44]. Microorganisms of the genus *Bacillus* are probably the most important bacterial source of proteases, as they are capable of producing high amounts of neutral and alkaline proteolytic enzymes with remarkable properties, such as high stability at extreme temperatures, pH, organic solvents, detergents, and oxidizing compounds [66]. The proteases (subtilisin and glutamyl endopeptidase) of *B. pumilus* have been shown to degrade polymeric extracellular components and significantly eradicate the biofilm generated by *Serratia marcescens*, known as an opportunistic pathogen responsible for several hospital-acquired infections [67]. Although enzymes have been widely studied in science, more research is still needed regarding their effects in the human body.

### 3.4. Bacteriocins

Bacteriocins are polypeptides and proteins that form pores in bacterial membranes and inhibit cell wall synthesis [45]. Currently, there are five classes of bacteriocins, as follows:Class I: Small proteolytic and heat-resistant peptides substantially modified by transcriptionally specific enzymes. Examples: lantibiotics (nisin), sactipeptide, and loop peptides [68].Class II: Divided into four subtypes that are (1) pediocin-like, (2) two peptides, (3) circular, and (4) linear, not pediocin-like. They comprise small peptides resistant to temperature and pH [68].Class III: Large thermolabile peptides (>30 kDa) with complex activity and structure. This group includes helveticin, acidophylline, and lactacins (A and B) [69].Class IV: It consists of complex proteins conjugated with lipids or carbohydrates. Examples include lactocin S (glycoprotein) and mesenterocin (lipoprotein) [68,69].Class V: Peptides with circular structures without post-translational modifications, including enterokine AS-48 and gasericin A [69].

Nisin is one of the most studied and used bacteriocins in food science and industry. This bacteriocin is produced by *Lactococcus. lactis* subsp. *lactis* and is approved by the FDA as a Generally Recognized as Safe Additive (“GRAS”) [70]. The antimicrobial spectrum of nisin is broad and can inhibit the growth of *Staphylococcus aureus*, *Cutibacterium acnes, Mycobacterium smegmatis*, and strains of *Bacillus, Clostridium*, *Enterococcus, Mycobacterium*, and *Streptococcus* [71]. Furthermore, bacteriocins from *Enterococcus faecalis* and *L. casei* have been shown to be effective against urogenital, intestinal, and antibiotic-resistant pathogens [72,73].

### 3.5. Exopolysaccharides (EPSs)

EPSs are biopolymers produced by microorganisms during their growth. They vary according to the degree of branching from linear to highly branched molecules [74]. EPSs are divided into homopolysaccharides and heteropolysaccharides. Homopolysaccharides are composed of monosaccharides identical in their structure; for example, starch and cellulose. In comparison, heteropolysaccharides consist of different monosaccharides such as xanthan gum, pectin, and galactomannans, among others [75]. The probiotics that have been studied reagrding EPSs belong to the genus *Lactobacillus*. In a study by Xu et al. [76], they presented that EPSs produced by *L. buchneri* TCP016 (doses of 200–800 mg) were able to reduce liver damage by regulating the intestinal microbiota, since EPSs reduced the growth of *Helicobacteraceae, Lachnospiraceae,* and *Enterobacteriaceae*, but increased the proliferation of *Lactobacillus*, *Rikenellaceae*, *Bacteroidaceae*, and *Prevotellaceae.* In another study, Wang et al. [77] studied the EPS from *L. fermentum* S1 by applying doses of 1–4 mg/mL; these EPS showed potent antioxidant activities and prevented the formation of biofilms of *E. coli* and *S. aureus.* In addition, EPS from *L. casei* SB27 showed an increase in the concentration of these compounds (600 μg/mL), and antitumor effects in vitro were increased [78]. EPSs have demonstrated various health effects such as antioxidants, cholesterol-lowering effects, immunity, anti-aging, modulation of the intestinal microbiota, and antitumor [76,77,78,79].

### 3.6. Teichoic Acids

According to Van der Es et al. [80], teichoic acids are composed of anionic glycopolymers (ribitol) with repeated units of polyols attached to phodiester. There are two types of teichoic acids: Lipoteichoic Acids (LTAs), which are anchored to the membrane by a glycolipid, and Wall Teichoic Acids (WTAs), which are covalently bound to peptidoglycan [81]. Teichoic acids have important roles in bacteria, such as determining cell shape, regulating cell division, and providing vital metabolic aspects for cell physiology, and may confer pathogenesis and antibiotic resistance of Gram-positive bacteria [45]. The most studied teichoic acids in science are LTAs, due to their immune, antitumor, and antioxidant functional properties [82]. For example, several studies have shown that LTAs from *L. delbrueckii*, *L. sakei*, *L. rhamnosus* GG, and *L. plantarum* K8 have potential anti-inflammatory effects [83,84,85]. Wang et al. [86] studied the LTAs of *L. paracasei* D3–5 using a dose of 200 mg /kg per day. Their results showed that the compounds benefited the physical, cognitive, and anti-inflammatory functions in mice.

### 3.7. Vitamins

Vitamins are essential in the diet in small amounts to facilitate various body biological processes. LeBlanc et al. [87] noted that these compounds are important in regulating biochemical reactions in the cell, since some participate as precursors of intracellular coenzymes. Currently, thirteen vitamins are known as essential to human health, and these are classified as fat-soluble (vitamins A, D, E, and K) and water-soluble (vitamin C and vitamins of the B complex) [87]. The human body cannot synthesize most vitamins; therefore, consuming them externally through food is an excellent alternative. Bacterial probiotics can synthesize a large amount of vitamins, such as vitamin B1 (thiamine), vitamin B2 (riboflavin), vitamin B9 (folate), vitamin B12 (cobalamin), and vitamin K [44,45]. As can be seen, the B complex vitamins are the most produced by bacterial probiotics; production is important, since the B vitamins act in synergy in the body’s homeostasis by participating in metabolic processes, such as energy generation and the synthesis of red blood cells [87]. Deptula et al. [88] noted that vitamin B12 produced through microbial fermentation is preferable to chemical synthesis. Nataraj et al. [45] argued that using microorganisms in vitamin production is more economically feasible than fortifying with chemically synthesized pseudo-vitamins. Bacterial probiotics such as *L. brevis, L. fermentum*, *L*. *reuteri*, and *L. salivarius* have complete genes (ribA, ribB, ribG, and ribH) for riboflavin synthesis [89]. Bardosono et al. [90] conducted a clinical trial consisting of *Bif*. *animalis* subsp. *lactis* HNO19 supplementation in pregnant women. Their results showed that probiotic supplementation increased the concentration of vitamin B6 in the blood and vitamin B12 in the second and third trimesters, respectively. Therefore, consuming probiotics or foods with probiotics is vital to staying healthy, as vitamins can also be synthesized in the digestive tract.

### 3.8. Biosurfactants

Biosurfactants represent a wide diversity of polymers synthesized during the early and late logarithmic stationary phases of a microorganism’s growth cycle [91]. Biosurfactants consist of glycolipids, lipopeptides, phospholipids, neutral lipids, protein-polysaccharide complexes, and free fatty acids [45]. The species of *Lactobacillus* are the most studied concerning biosurfactants. The biosurfactants (methylpentadecanoic acid and eicosanoic acid) of *L. jensenii* P6A and *L. gasseri P6* have shown antimicrobial properties against several pathogens, such as *E. coli*, *Candida albicans*, *Staphylococcus saprophyticus*, *Enterobacter aerogenes*, and *Klebsiella pneumonia* [92]. Merghni et al. [93] showed that the biosurfactants of *L. casei* ATCC 393 have antioxidant and antimicrobial properties against *S. aureus* (present in the oral cavity), suggesting that biosurfactants could have a significant role in the prevention of oral diseases. Fracchia et al. [94] noted that biosurfactants exhibit a wide variety of properties such as emulsion stabilization, anti-adhesion capacity, anti-biofilm, anticancer, antiviral, immunological, and antimicrobial.

## 4. Health Benefits of Food Added with Probiotic Bacteria

This section presents the most recent studies regarding the health benefits and consumption of foods with probiotics (fermented or microencapsulated) and some postbiotics. Table 1 shows the scientific studies (in vitro and in vivo) that have been reported concerning the health benefits that can be obtained by consuming foods with probiotics and postbiotics, as well as the probiotics that have been studied and the main obtained results.

### 4.1. Benefits on Gut and Gastrointestinal Tract

The intestinal tract is a very important organ for the digestive and immune systems; a weak or unhealthy intestinal tract affects the metabolism, immunity, and mental condition [138]. The gut microbiota is a consortium of a wide variety of microorganisms that provide energy and nutrition, and protect the integrity of the gut against pathogenic bacteria [139]. Therefore, regulation of the gut microbiota is vital to prevent gastrointestinal distress such as constipation, diarrhea, and intestinal ulcers. Probiotics are mainly related to the intestinal tract, as their consumption improves the immune system by regulating the intestinal flora and favors the production of metabolites [140]. Gao et al. [138] presented a bibliographic compilation showing how the metabolites produced by probiotics can regulate the intestinal microbiota. EPS, SCFA, and bacteriocins are the main compounds that regulate the intestinal microbiota. For example, SCFAs (acetic, propionic, and butyric acid) regulate the pH of the intestine and stimulate the mucous production of epithelial cells, which can prevent the adhesion and proliferation of pathogenic bacteria [57,140]. Daliri et al. [141] proposed patterns of probiotics’ mechanisms of action in the intestinal micro-ecosystem: modulation of endogenous colonies, protection barrier of the intestinal tract, homeostasis maintenance, regulation of the immune system, and influence on the vagal afferents.

In foods with probiotics, their consumption has been evaluated to understand their effect on the intestinal tract and microbiota. For instance, Liu et al. [95] studied yogurt consumption with *Lc. lactis*, *L. plantarum*, and *L. casei* compared with commercial yogurt (*L. bulgaricus* and *St. thermophilus*) in male rods (*n* = 144) with constipation. Their results showed that yogurt consumption changed the microbiota, giving higher percentages for the phylum *Bacteroidetes* and lower percentages for the phylum *Firmicutes*. In addition, consuming both yogurt samples significantly reduced mice defecation by relieving loperamide-induced constipation. Likewise, in rodent models, it has been shown in several studies that milk consumption fermented with *L. plantarum B7*, *L. rhamnosus* S1K3, and *L. paracasei* ST11 can reduce the population of pathogenic bacteria, such as *Salmonella enterica* serovar Typhimurium [97,98,99]. In clinical trials, Kato-Katoaka et al. [125] studied the effects of fermented milk with *L. casei* Shirota on abdominal dysfunction in 47 healthy medical students during eight weeks of intervention. In their results, it was observed that the beverage with *L. casei* Shirota was able to reduce stress-induced abdominal dysfunction by regulating the microbiota, since the group that consumed the fermented beverage significantly reduced the percentage of *Bacteroidaceae* species compared to the placebo group. In a systematic review by Donovan and Rao [142], several studies were identified that showed that yogurt consumption had a significant effect on infectious diarrhea in healthy and malnourished children. The consumption of milk fermented with *L. paracasei* and licorice (*Glycyrrhiza glabra*) demonstrated effects against *Helicobacter pylori* infection (causing gastric ulcer) in a clinical study by Yoon et al. [143]. The results showed that patients who consumed the product reduced the histological inflammation produced by *H. pylori* for eight weeks of intervention. In addition, postbiotic food, such as fermented milk with heat-inactivated *L. gasseri* CP2305, has been reported to regulate bowel function in people with tendencies to constipate (*n* = 39) [103]. The consumption of foods with probiotics improves the intestinal wellbeing of people. However, we still need to understand more about probiotics’ competence, mechanisms of action, and metabolites during their intestinal tract passage.

### 4.2. Antioxidant Properties

The study of antioxidant properties has mainly focused on reducing the effects derived from oxidative stress. Oxidative stress is the intracellular state in which oxygen radical levels are elevated, so this increase in radicals can damage lipids, proteins, and DNA [144]. These compounds are commonly referred to as reactive oxygen species (ROS), comprising superoxide anion radicals, hydroxyl radicals, and hydrogen peroxide [145]. ROS at high intracellular levels are detrimental to different biological processes involving stem cell exhaustion, tumorigenesis, autoimmunity (immune response against self-antigens), and accelerated senescence [144].

Naturally, living organisms have various defenses against oxidative damage, such as enzymatic (superoxide dismutase (SOD), glutathione peroxidase (GPx), glutathione reductase (GR)), non-enzymatic (glutathione, thioredoxin, vitamin C, and vitamin E) and repair systems of the organism [146]. Despite possessing these defenses, it is not enough to avoid the effects of oxidative stress. Hence, one way to combat it is through diet, which involves the consumption of foods with antioxidant compounds. Several studies (Table 1) have shown that consuming foods with probiotics induces antioxidant effects on the human body. For instance, Shori et al. [105] studied various cashew-plant-based milk yogurt samples with *L. rhamnosus* ATCC 53103, *L*. *casei* ATCC 393, and *L. plantarum* ATCC 14917. The antioxidant activities by 2,2-diphenyl-1-picrylhydrazyl (DDPH) assay, capacity chelating ferrous ions (FIC), and potential ferric reducing antioxidant (FRAP) were evaluated. Milk-based cashew samples from *L. rhamnosus* ATCC 53103, *L. casei* ATCC 393, and *L. plantarum* ATCC 14917 showed higher values in antioxidant activity tests than the control yogurt made with cow’s milk and commercial cultures (*St*. *thermophilus* St1342 and *L. delbrueckii* ssp. *lactis* ATCC 7830). The sample with *L. rhamnosus* ATCC 53103 had the highest antioxidant capacity in the DPPH, FIC, and FRAP tests. However, it is worth mentioning that the sample with *L. plantarum* ATCC 14917 had a considerable increase in antioxidant effect in the FIC and FRAP tests at 21 days of storage. In another example, Duru et al. [106] developed a fermented oat product with *L. acidophilus* DSM 13241 and *St. thermophilus* DSM 15957; they formulated three samples that varied in the amount of isoflavones 0.32, 0.16 mg/mL, and without addition for A, B, and C, respectively. The results showed that antioxidant activity (DPPH assay) increased with the addition of isoflavones, but this was reduced according to longer storage time (up to 28 days). Duru et al. [106] mentioned that adding isoflavones to fermented oats improved the microbial count of bacterial probiotics but did not improve their viability during product storage. Antioxidant properties have also been evaluated in animal models, such as mice. One piece of evidence is the study by Yoon et al. [102], in which a probiotic yogurt was prepared with *L. acidophilus*, *Bif. lactis*, *St. thermophilus*, and *L. delbrueckii* subsp. *bulgaricus* fermented at low temperatures (22 °C for 27 h) and conventionally fermented (37 °C for 12 h). Evaluations of antioxidant activities (DPPH assay, 2,20-azinobis (3-ethylbenzothiazoline-6-sulfonic acid) (ABTS) cation assay, and FRAP assay) were applied to mice induced with colitis. The results showed that slow fermentation of probiotic yogurt obtained a higher antioxidant response than rapid fermentation. Yoon et al. [102] argued the reason for this difference is that probiotics had a better absorption of the substrate, given its solubility and diffusivity.

Furthermore, antioxidant properties have been analyzed in foods with encapsulated bacterial probiotics. One study evaluated the antioxidant activities (DPPH assay, 2,20-azinobis (3-ethylbenzothiazoline-6-sulfonic acid) (ABTS) cation assay, oxygen radical absorbance capacity (ORAC) assay, and photochemiluminescence (PCL) assay) of two snack bars made with chickpeas and green lentils with *L. plantarum* microencapsulated (specifically in the chocolate coating) [107]. The results showed that the snack bar made with chickpeas presented greater antioxidant activity than the bar made with green lentils. It was also mentioned that the total content of polyphenols (TPC) influenced the values of antioxidant activities (ORAC, DPPH, and ABTS). Postbiotic foods have also been reported to exhibit antioxidant activities. For example, Gholamhosseinpour and Hashemi [108] evaluated the antioxidant activity of fermented milk with ultrasound-inactivated *L. plantarum* AF1 to stop fermentation and to not affect thermolabile compounds. In their results, fermentation was observed to increase antioxidant activity (ABTS assay and DPPH assay), and ultrasound significantly improved this activity, especially the 15-min treatment. They argued that this may be caused by cell lysis of bacteria (release of intracellular compounds), the modification of bacterial extracellular metabolites, and hydrolyzed constituents of milk [108]. Certain compounds derived from probiotic fermentation have been considered antioxidant agents, which include organic acids (SCFA), lactate, 3-phenyllactate, indole-3-lactate, β-hydroxybutyrate, and γ-aminobutyrate, among others [109].

As has been observed, in vitro or in vivo tests show that foods with bacterial probiotics can provide antioxidant activities. However, the best way to verify these activities is through clinical studies focused on oxidative stress. For example, the daily consumption of 200 mL of soy milk with *L. plantarum* A7 has been evaluated in patients with diabetic kidney disease [110]. The results showed that consumption of probiotic soy milk had a slight effect on the oxidative stress of patients, as oxidized glutathione concentration (ROS) levels were reduced from intake of the probiotic product compared to soy milk without the added probiotic.

Unfortunately, sometimes no improvements or effects in consumption are observed. For example, Bahmani et al. [111] studied the effects of the consumption of symbiotic bread, which contained *L. sporogenes* and inulin (7 g of inulin per 100 g sample). The results showed that consumption did not have an effect or increased the antioxidant capacity of the plasma. Therefore, the authors argued that a more extended study time was necessary, since the time was short (8 weeks) and to apply more antioxidant activity tests, such as glutathione peroxidase and superoxide dismutase [111]. Another study evaluated the effect against oxidative stress from the supply of probiotic yogurt with *L. acidophilus* LA5 and *Bif. lactis* BB12 in pregnant women over nine weeks [112]. The results showed that the consumption of probiotic yogurt did not have significant differences in terms of oxidative stress parameters (total antioxidant activity (TAC), plasma glutathione (GSH), erythrocyte glutathione peroxidase erythrocyte glutathione peroxidase (GR), and serum 8-oxo-7,8-dihydroguanine serum 8-oxo-7,8-dihydroguanine (8-oxo-G)) compared to the control yogurt. Similarly, the authors had the same comments regarding the intervention time, that it was very short, and also referred to the small budget of the clinical study. Therefore, research still needs to be carried out on antioxidant activities, which require more clinical studies evaluating the consumption of foods with probiotics.

### 4.3. Anti-Inflammatory Properties

According to Plaza-Díaz et al. [147], the consumption of probiotics may provide anti-inflammatory effects that help relieve chronic intestinal diseases, inflammatory bowel diseases (IBDs), necrotizing enterocolitis (NEC), and malabsorption; being the syndromes with the most clinical manifestations. The term IBD describes four pathologies as ulcerative colitis (UC), Crohn’s disease (CD), pouchitis, and microscopic colitis. Table 1 shows clinical studies of probiotic foods with anti-inflammatory effects; there are few because most have been studied with dairy foods, such as yogurt. For example, Mazani et al. [113] conducted a clinical study involving the effect of consuming a probiotic yogurt with *Lactobacillus* spp. in young women (*n* = 27) after exercise. Results showed that consumption after exercise of probiotic yogurt for two weeks significantly decreased (*p* < 0.05) serum tumor necrosis factor-alpha (TNF-α) and biochemical factors matrix metalloproteinase 2 (MMP2), matrix metalloproteinase 9 (MMP9), and malondialdehyde (MDA). In contrast, the control group showed no effect on biochemical factors or anti-inflammatory tests. Another clinical study evaluated the effect of consuming sheep’s milk yogurt with probiotic cultures (*St. thermophilus* and *L. bulgaricus*) and commercial yogurt prepared with cow’s milk [114]. The study consisted of thirty participants and lasted approximately five weeks; three yogurt samples were evaluated: whole sheep’s milk, semi-skimmed sheep’s milk, and cow’s milk. Overall, the results showed that there were no significant differences between the samples in terms of inflammatory serum tests (IL-8, TNF-α, and IL-10), but there were decreases in proinflammatory expressions of tumor necrosis factor-alpha (TNF-α) and cytokine IL-8, and increased expression of the anti-inflammatory cytokine IL-10 between the beginning and end of the intervention. Redondo et al. [114] argued that the size of the sample or participants, as well as the time of study, can be considerable influences, so it is recommended to have a large number of participants, at least 50, and a major period of consumption, in addition to the fact that clinical trials are complex because each participant is naturally different. It is not easy to control the diet of each participant.

The study by Lee et al. [115] evaluated the effect of daily consumption (for 12 weeks) of yogurt containing *L. paracasei* (*L. casei* 431^®^) and *Bif. lactis* (BB-12^®^) and postbiotic *L. plantarum* nF1 (inactivated with heat treatment) in 200 individuals over 60 years old. The results showed that the consumption (n = 100) of probiotic and postbiotic yogurt increased cytokine (IL-12) activity, which is linked to inflammatory bowel diseases. Other probiotic foods have only been studied with Kimchi [109] and *Inula britannica* (East Asian medicinal herb) [116], but studies of these foods have only used in vitro tests to analyze the anti-inflammatory properties. For example, Kim et al. [109] used the gas chromatography technique to investigate metabolites generated during kimchi fermentation with *L. plantarum* PMO 08. Their results identified around 78 metabolites, some of which have been closely related to anti-inflammatory activities, such as lactate, indole-3-lactate, β-hydroxybutyrate, γ-aminobutyrate, and glycerol. Based on this molecular identification, they presented that kimchi fermented with *L. plantarum* PMO 08 has anti-inflammatory activities, since in vitro tests showed a significant reduction in levels of proinflammatory cytokine expressions (IL-6 y TNF-α). Despite the breakthrough and the studies reported in the literature regarding anti-inflammatory properties, the mechanisms of action are still unclear and not fully understood, as it has been reported that the activity during immunomodulation is highly specific to the microbial strain and depends on the health status of the host [148].

### 4.4. Anti-Cancer Properties

Cancer is a disease of high importance in medicine, as it is the second cause of mortality worldwide [149]. The World Health Organization (WHO) [150] reported around 18.1 million new cases in 2018, and this number is expected to rise to 29.4 million by 2040. Colorectal, prostate, lung, stomach, liver, and breast cancers are the highest incidences of this fatal disease [151]. Cancer is a disease caused by DNA repair deficiency or mutations produced during DNA replication, and exposure to the environment, habits of daily life, and genetic inheritance of individuals are the main risk factors [152,153]. Cancer is an uncontrolled cell division usually detected in the late stages of its progression. Once cancer cells spread, they eventually need multiple stages for tumor development to become malignant [154].

Today, cancer treatment is based on the supply of multiple drugs and chemotherapy, which damage not only cancer cells but also healthy cells, causing a certain resistance that can affect future treatments of the patient [155]. Hence, consuming foods with probiotics has been suggested for treating and preventing cancer, as several studies have shown anticancer properties (Table 1). Fatahi et al. [117] evaluated the anticancer activity of kefir (considered a probiotic food), in which different concentrations of kefir extracts interacted with a U87 glioblastoma cell line. Furthermore, anticancer activity was evaluated with the cytotoxic effect on cancer cells using an MTT test. The results found by Fatahi et al. [117] showed that a higher concentration (20 mg/mL) of kefir and fermentation time (48 h) managed to inhibit most malignant cells (83%). They argued that this inhibition might be caused by the interaction of metabolites produced during fermentation and cancer cells that can interfere with regulating proliferation, cell distinction, apoptosis, metastasis, and angiogenesis.

Another example is the evaluation of the anticancer activity of milk fermented with *L. casei* ATCC 393, which was carried out by Abdel-Hamid et al. [118]. For anticancer activity, it exposed the aqueous extract peptide and its filtrate (<2 kDa) with cell lines of carcinoma MCF-7 and Caco-2 obtained as a response to the anti-proliferative activity using MTT assay. The results showed that peptides with lower molecular weight (<2 kDa) obtained higher values of anti-proliferative activity than the unfiltered aqueous peptide extract in both cell lines, increasing the anticancer properties of cow’s milk fermented with *L. casei* ATCC 393 [118]. Similarly, it has been shown that peptides from the fermentation of *Lc. lactis* KX881782 and *L. acidophilus* DSM9126 in camel’s milk have more cellular inhibition of Caco-2, MCF-7, and HELA than in the fermentation of cow’s milk [119]. It has been argued that peptides from probiotic fermentation, due to their hydrophobicity, amphipathic property, secondary membrane structure, net charge, and oligomerization capacity, are the most important contributors to anticancer activities [156].

In clinical studies, a probiotic drink with *L. casei* Shirota incorporated with soy isoflavone in healthy people (*n* = 309) has been evaluated to prevent breast cancer [121]. The results were obtained through clinical questionnaires, which showed a biological interaction between people who consumed the probiotic drink and soy isoflavones. It was observed that the consumption of this product could prevent breast cancer, but it was impossible to identify the amount or daily dose necessary to prevent this disease. Furthermore, Toi et al. [121] suggested that lifestyle and diet must be modified for cancer prevention, including consuming their probiotic drink with *L. casei* Shirota and soy isoflavone. Similarly, Pala et al. [122] studied the protective effect against colorectal cancer (CRC) through the consumption of commercial yogurt (*St. thermophilus* and *L. delbrueckii* subsp. *bulgaricus*) in a population of 45,241 participants over 12 years of intervention. Again, clinical questionnaires were used to obtain results. During that period, 289 volunteers were diagnosed with CRC. In the results, it was observed that yogurt consumption was inversely associated with the risk of contracting CRC, and the protective effect of yogurt was evident throughout the population.

Furthermore, it was observed that the protective effect was stronger in men than in women. Clinical studies on anticancer properties have been very few and have been based on questionnaires. The results do not show biological or biochemical evidence that could verify whether there certainly is an anticancer effect in human models. More clinical studies are needed to show that the consumption of probiotic foods helps treat and prevent cancer.

### 4.5. Psychobiotic Effects

Long-term consequences that affect today’s society can be associated as a result of the pandemic generated by the COVID-19 virus. Due to quarantine-induced isolation, the sudden stoppage of the daily lives of millions of people led to increases in loneliness, anxiety, depression, insomnia, the harmful use of alcohol and drugs, and self-harm or suicidal behavior [157,158,159]. For this reason, possible treatments to alleviate mental disorders caused by the pandemic have been investigated, and consuming probiotics or foods could be a viable solution. Recent research has argued that gut microbiota affects gut regulation and influences mental health through the gut-brain axis [124,125]. Although the term “psychobiotics” originated in 2013 [160], it has gained more potency in recent years due to the current situation. Psychobiotics refers to the psychological potential of molecules produced by probiotic bacteria and refers to certain probiotics that include species such as *St. thermophilus, Bif. animalis*, *Bif. bifidum*, *Bif. longum*, *L. bulgaricus*, *Lc. lactis*, *L. acidophilus*, *L. plantarum*, *L. reuteri*, *L paracasei*, *L. helveticus*, *L. rhamnosus*, *B. coagulans*, and *Clostridium butyricum*, among others. [161]. These bacterial probiotics must be able to produce neuroactive substances, such as γ-aminobutyric acid (GABA) and serotonin [161].

Even the mechanism of action of psychobiotics is not fully understood. Still, it is proposed that the intestinal microbiota communicates with the brain, and vice versa, through the central, autonomic, and enteric nervous systems, as well as the hypothalamic-pituitary-adrenal (HPA) axis [162]. Communication between the gut and the brain must be efficient. Hence, several pathways are concentrated in the nervous system and involve cytokines released by mucosal immune cells and hormones produced by endocrine cells or through the vagus nerve. In addition, the synthesis of neurotransmitters, neurochemicals, and metabolites is important for proper signaling in the brain concerning body stress [161]. Such is the case of serotonin, of which Yano et al. [163] noted that more than 90% of this compound in the body is produced in the intestine and the gut-brain axis, which is vital for the regulation of activation receptors located in enterocytes, enteric neurons, and immune system cells. The main mechanism suggests that when the production of neurotransmitters in the intestine is increased, it causes a decrease in tryptophan in the blood plasma, which generates an increase in the transmitter molecules of the brain and can improve neuropsychiatric parameters such as sleep, appetite, mood, and cognition [164]. Another mechanism proposes that metabolites, such as SCFAs, as they are involved in decreasing pro-inflammatory cytokines, may positively affect the treatment of stress and depression [161,165]. Butyrate is the SCFA that has been the most studied, since bacterial probiotics such as *Bif. breve* M2CF22M7, *Bif. infantis* E41 and *C. butyricum* have demonstrated antidepressant properties in rodents [166,167].

Concerning foods with probiotics, there have been several clinical studies that have evaluated the psychobiotic effects, as shown in Table 1. For instance, Chung et al. [123] studied the psychobiotic effect of fermented milk tablets with *L. helveticus* IDCC3801 in healthy elderly people aged 60 to 75 years old for 12 weeks. The results showed that fermented milk tablets improved patients’ cognitive functions compared to the placebo group. Similarly, it has been shown that the intake of fermented milk with *L. helveticus* CM4 with lactononadecapeptide in 61 healthy middle-aged people (50–70 years old) for eight weeks [124]. In the results of the cognitive questionnaires, cognitive metabolism, attention, and memory were demonstrated compared to the placebo group. Concerning serotonin production, Kato-Katoaka et al. [125] investigated the physiological effects and physical stress of consuming fermented milk with *L. casei* Shirota in 47 medical students undergoing academic examinations for eight weeks. It was possible to demonstrate that the probiotic food consumption significantly increased the levels of fecal serotonin compared to the placebo group. In addition, consumption significantly reduced the stress of students. Other fermented products with probiotics (mainly of the genus *Lactobacillus* and *Bifidobacterium*), such as fermented algae (*Laminaria japonica*) [126], fermented soy [127], kefir [168], and yogurt [128,129], showed psychobiotic effects (Table 1), such as antidepressant, anti-stress, enhanced emotions in women, cognitive enhancement and intelligence, and effects against Alzheimer’s syndrome and anxiety.

Furthermore, dairy foods have been studied with postbiotics, such as the study by Nishida et al. [130], which evaluated the psychobiotic effects of postbiotics from *L. gasseri* CP2305, developing a fermented milk beverage. The study focused on healthy students (21 men and 11 women) and the intervention time was five weeks with doses of 190 g per day. To promote a state of stress, the study was conducted when students were taking a cadaveric dissection course. It showed that dairy beverages with inactivated LAB improved sleep quality in men more than in women. Furthermore, sleep time and latency increased and decreased in men, respectively. Science has made remarkable findings regarding the psychobiotic effects of consuming foods with probiotics and postbiotics. However, several researchers have concluded that the mechanisms of action and the relationship between the microbiota and the brain still need further investigation.

### 4.6. Against Heart Diseases and Obesity Reduction

Heart disease is a major concern for the global health system, as one in three people have high cholesterol levels in developed economies [169]. We can also add it to the increase in obesity worldwide owing to a diet high in fatty foods, such as junk foods. Obesity is an extreme state of fat accumulation in the body, closely related to heart and kidney illnesses [169]. Table 1 presents the studies that have been conducted on heart disease and consuming foods with bacterial probiotics. In a study conducted using rats, it was observed that fermented milk incorporated with *L. fermentum* MTCC significantly reduced levels of low-density lipoprotein cholesterol, serum total cholesterol, liver lipids, coronary artery risk index, and TNF-α and IL-6 mRNA expression of the liver [131]. Beltran-Barrientos et al. [132] conducted a clinical study evaluating the effect of blood pressure reduction on consuming fermented milk with *Lc. lactis* NRRL B-50571 in 18 patients with a tendency to hypertension with an intervention time of 5 weeks. The results revealed that daily consumption of the probiotic product reduced patients’ systolic and diastolic blood pressure and decreased serum levels of triglycerides, total cholesterol, and low-density lipoproteins. However, the consumption of yogurt with *L. acidophilus* La5 and *Bif. lactis* Bb12 has no observed effects against heart disease; the authors argued that this lack of effects was due to the short treatment time (10 weeks) [133]. Regarding anti-obesity effects, it has been suggested that a diet based on probiotic interventions is more effective than drug delivery [169]. Razmpoosh et al. [134] evaluated the effects of the yogurt condensed with *L. acidophilus* La5 and *Bif. lactis* Bb12 consumption in 70 overweight or obese women over an intervention time of 8 weeks. The results showed that the intervention group had a greater reduction in triglyceride and cholesterol levels than the placebo group. In addition, fat percentage, body mass index, and waist circumference were significantly more reduced in the intervention group than in the placebo. Similarly, the anti-obesity effects of milk fermented with *L. casei* Shirota have been reported; its consumption can decrease glycol-albumins and low-density lipoproteins and reduce adipose tissue [135].

### 4.7. Antiviral Properties

With the recent pandemic caused by COVID-19, research into antiviral properties has gained a lot of attention as the efforts of science and many researchers, as well as vaccines, were obtained in a very short time. Despite the existence of vaccines and drug tests, some people show limited efficiency, so the consumption of probiotics is a viable option to increase defenses in people with a compromised immune system [170]. Muhialdin et al. [171] showed the antiviral activity of probiotics isolated from fermented foods from various studies, most of which had antiviral properties against influenza viruses and enteroviruses; in vitro and in vivo studies were conducted in animal models. As shown in Table 1, few studies have been conducted mainly focusing on the antiviral properties of foods with probiotics or postbiotics. Kim et al. [136] demonstrated the antiviral effects of yogurt with thermally inactivated *L. plantarum* nF1 (postbiotic) in mice. Their results showed that postbiotic yogurt raised the cytokine Natural Killers (NK) expression, which is closely related to the modulation of the immune system and the primary defense against viral infections.

Similarly, it has been reported that *L. delbrueckii* subsp. *bulgaricus* OLL1073R-1 in fermented milk may provide antiviral effects against influenza A [137]. It has been noted that metabolites like proteinase inhibitors and bacteriocins are the main causes of the antiviral activity of probiotics [172]. It is necessary to conduct clinical studies that evaluate this property, as they could help fight viruses such as COVID-19, or be used for the prevention of stationary viruses (Influenza) or new viruses that appear.

## 5. Probiotics and Postbiotics in the Food Industry

As previously mentioned, probiotics can be presented in various forms for consumption. Therefore, the food industry has a great opportunity to develop products with very high bioactive potential as functional foods, since probiotics and postbiotics can be added to the formulation or preparation of the product. In probiotics, there is a great diversity of products that contain them, most of which are dairy. However, as Reque and Brandelli [173] noted, there are sectors of the population that demand non-dairy products due to either health issues (lactose or milk protein intolerance) or lifestyles (vegetarians or vegans). With the help of microencapsulation of probiotics, products can be designed for people seeking alternatives to dairy products. There are food alternatives with microencapsulated probiotics, such as fruit-based drinks, meat products, desserts, chocolates, and bakery products [16]. While, for lifestyles, more research is required for those that only consume vegetables. Specifically, it is necessary to identify which probiotics are suitable, since their origin must be from vegetable sources. In addition, probiotics help cover the nutritional needs of vegetarians/vegans, such as the production of complex B vitamins or improving the biodigestibility of proteins in the food [174]. In addition, various antinutritional compounds of vegetables are delayed by probiotic growth.

In the same way, postbiotics are an excellent alternative since they are more stable and do not need strict storage, handling, and transport conditions, such as a cold chain. Unfortunately, there have been few studies in which postbiotics are incorporated into foods, but foods such as yogurt and whey-based sports drinks contain inactivated LAB bacteria. A few examples include *L. bulgaricus*, *St. thermophilus*, *L. acidophilus*, *L. gasseri* CP2305, and *L. casei* 01, which have already been carried out [103,130,175,176].

A very interesting point regarding probiotics and postbiotics is that they not only provide benefits to the health of the consumer, but they can also participate in food safety because potential applications of postbiotics in food safety (mainly LAB) have been discovered. These include the bio-preservation of dairy, meat, vegetable, and bakery foods, and the production of antimicrobial compounds such as organic acids and/or bacteriocins. This is in addition to its use in the development of bioactive and edible packaging; the prevention and control of biofilms in food machinery; and the reduction and degradation of contaminants, such as biogenic amines, pesticides, and mycotoxins [177]. More research on postbiotics is needed to be able to introduce them into the food industry, since their interaction with food components could produce defects in sensory properties, such as taste, odor, or appearance. Similarly, in products with microencapsulated probiotics, there may be defects that could occur in the shelf-life of the food due to fermentations after processing. Techniques are needed that keep the probiotics partially inactivated or latent. This research is critical to support future regulations and applications with either postbiotics or microencapsulated probiotics incorporated into foods.

## 6. Conclusions and Future Perspectives

Bacterial probiotics can be incorporated into foods in various ways, whether they are involved in the product’s fermentation, microencapsulated for better bioactivity delivery, or inactivated for better stability in the food product. The fermentation of bacterial probiotics generates a great diversity of metabolites, which have been investigated as demonstrating benefits to human health. Nonetheless, most studies concerning metabolites have been in vitro and in vivo (mostly in rodents), which cannot completely assure that the same benefits would occur in the human organism, so it is therefore necessary to investigate in humans. On the other hand, a large amount of scientific evidence has been presented regarding the consumption of foods with bacterial probiotics. These foods provide health benefits such as the regulation of the microbiota, help relieve gastrointestinal disorders and heart diseases, and have antioxidant, anticancer, anti-inflammatory, psychobiotic, anti-obesity, and antiviral capacities.

Although clinical studies have been the primary sources of valuable information, there are still many loose ends regarding the mechanisms of action of bacterial probiotics and the reasons for alleviating so many ailments. Therefore, in the future, more research involving clinical studies is needed, since foods related to bacterial probiotics could help reduce the consumption of drugs and prevent certain diseases in conjunction with a healthy lifestyle and diet. Furthermore, more research is needed regarding microencapsulated and postbiotic foods, since most have been used in the application of fermented foods, especially in dairy products. However, as previously mentioned, foods with microencapsulated and postbiotic probiotics can be added to non-dairy foods. Therefore, this is a significant opportunity for the food industry and research within nutrition, medicine, and food science.

## Figures and Tables

**Figure 2 molecules-28-01230-f002:**
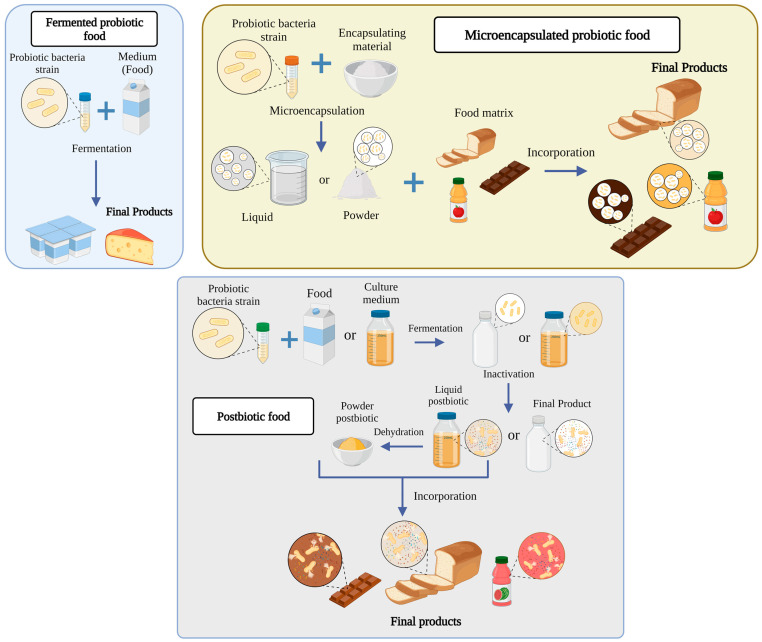
Types of food with probiotic bacteria.

**Figure 3 molecules-28-01230-f003:**
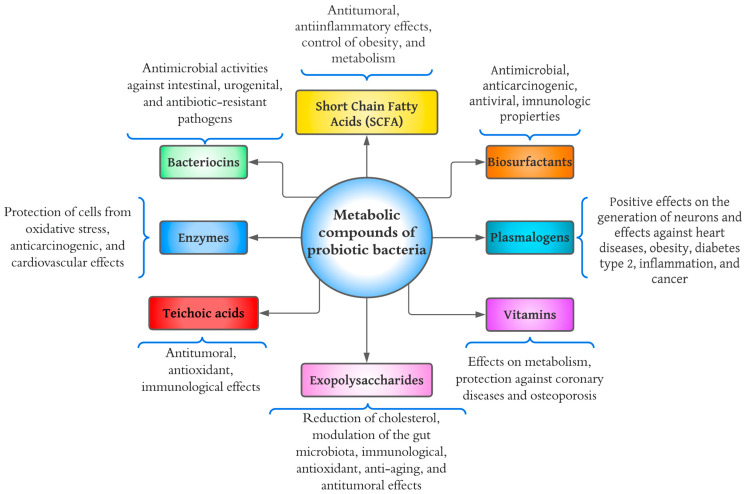
Diagram of health benefits of metabolic compounds of probiotic bacteria.

**Table 1 molecules-28-01230-t001:** Selected studies on the health benefits of consuming food with probiotic bacteria and postbiotics.

Probiotic Bacteria	Food Matrix	Type of Analysis	Main Results	Reference
*Lc. lactis, L. plantarum,* and *L. casei*	Yogurt	In vivo (mice models)	Regulation of gut microbiota, and relief of constipation	[95]
*L. plantarum* B7	Fermented cow’s milk	In vivo (mice models)	Antipathogenic activity of *Salmonella* spp.	[96]
*L. paracasei* NCC 2461				[97]
*L. rhamnosus* S1K3				[98]
*L. casei*	Fermented cow’s milk and gelled milk	Clinical trial	Reduction of diarrhea in healthy children	[99]
*L. delbrueckii* subsp. *bulgaricus* and *St. thermophilus*	Yogurt	Clinical trial	Reduction of diarrhea in unhealthy children	[100]
	Yogurt	Clinical trial		[101]
*L. paracasei*	Fermented cow’s milk	Clinical trial	Reduction of histological inflammation caused by *Helicobacter pylori*	[102]
*L. gasseri* CP2305 heat inactivated	Fermented cow’s milk	Clinical trial	Regulation of gut microbiota in constipation people	[103]
*L. plantarum, L. rhamnosus* GG, *L. acidophilus* and *L. casei* Shirota	Beverage of fruit juices	In vitro	Antipathogenic activity against *E. coli, Salmonella enteritidis, Shigella dysenteriae*, and *Shigella flexneri*	[104]
*L. rhamnosus* ATCC 53103, *L. casei* ATCC 393 and *L. plantarum* ATCC 14917	Cashews yogurt	In vitro	Antioxidant activity in DDPH, FIC, and FRAP assay	[105]
*L. acidophilus* DSM 13241 and *St. thermophilus* DSM 15957	Oat fermented	In vitro	Antioxidant activity in DDPH assay	[106]
*L. acidophilus*, *Bif. lactis*, *St. thermophilus*, and *L. delbrueckii* subsp. *bulgaricus*	Yogurt	In vivo (mice models)	Antioxidant activity in DDPH and ABTS assay	[102]
*L. plantarum* microencapsulated	Snack bar of lentils and chickpeas	In vitro	Antioxidant activity in DDPH, ABTS, ORAC, and PCL assay	[107]
*L. plantarum* AF1 inactivated	Fermented cow`s milk	In vitro	Antioxidant activity DDPH and ABTS assay	[108]
*L. plantarum* PMO 08	Kimchi	In vitro	Antioxidant and anti-inflammatory activities on RAW 264.7 cells	[109]
*L. plantarum* A7	Fermented soy milk	Clinical trial	Reduction of oxidative stress in people with diabetes	[110]
*L. sporogenes*	Symbiotic bread	Clinical trial	Without effects on oxidative stress	[111]
*L. acidophilus* LA5 and *Bif. lactis* BB12	Yogurt	Clinical trial	Without effects on oxidative stress	[112]
*Lactobacillus* spp.	Yogurt	Clinical trial	Reduction of proinflammatory expressions TNF-α and biochemical factors MMP2, MMP9, and MDA on serum blood	[113]
*St. thermophilus* and *L. delbrueckii* subsp. *bulgaricus*	Goat’s milk yogurt	Clinical trial	Reduction of proinflammatory expressions cytokines (IL-8 and TNF-α) and activation of anti-inflammatory cytokines (IL-10)	[114]
*L. paracasei* (*L. casei* 431^®^), *Bif. lactis* (BB-12^®^), and *L. plantarum nF1* inactivated	Yogurt	Clinical trial	Anti-inflammatory activities through the Increase of cytokine activity	[115]
*Weissella cibari*	Inula britanica (medicinal herb of Asia east)	In vitro	Inhibition of nitric oxide and proinflammatory cytokines by inhibiting NF-kB	[116]
Consortium of LAB and yeast	Kefir	In vitro	Inhibition of malign cells (cell lines U87 glioblastoma)	[117]
*L. casei* ATCC 393	Fermented cow’s milk	In vitro	Anti-proliferative cancer cells of peptide compounds in MCF-7 and Caco-2 line cells	[118]
*L. lactis* KX881782 and *L. acidophilus* DSM9126	Fermented camel’s milk	In vitro	Anti-proliferative cancer cells of peptide compounds in Caco-2, MCF-7, and HELA cells	[119]
*Lb. plantarum* LP3 and LU5	Fermented goat’s milk	In vitro	A positive effect of ultrasonication (60% amplitude) in enhancing anticancer activity	[120]
*L. casei* Shirota	Beverage with soy isoflavone	Clinical trial	Potential prevention of breast cancer in Japanese young women	[121]
*St. thermophilus* and *L. delbrueckii* subsp. *bulgaricus*	Commercial yogurt	Clinical trial	Protector effect against cancer	[122]
*L. helveticus* IDCC3801	Fermented cow’s milk tablets	Clinical trial	Enhance cognitive functions in healthy elderly people	[123]
*L. helveticus* CM4	Fermented cow’s milk	Clinical trial	Attention and memory functions were enhanced in healthy people	[124]
*L. casei* Shirota	Fermented cow’s milk	Clinical trial	Regulation of gut microbiota, an increase of serotonin in fecal samples, and stress reduction in medicine students	[125]
*L. brevis* BJ20	Fermented Laminaria japonica	Clinical trial	Reduction of degenerative effects (short-term memory) and physical function	[126]
*L. plantarum* C29	Fermented soy milk	Clinical trial	Enhance cognitive function in individuals with mild cognitive impairment	[127]
*L. acidophilus* LA5 and *Bif. lactis* BB12	Yogurt	Clinical trial	Stress reduction in petrochemical workers	[128]
*L. gasseri* SBT2055 and *Bif. longum* SBT2928	Yogurt	Clinical trial	Alleviating the stress of healthy people	[129]
*L. gasseri* CP2305 heat-inactivated	Fermented cow’s milk	Clinical trial	Enhance sleep quality, increase sleep time, and reduction of latency during sleep	[130]
*L. fermentum* MTCC	Fermented cow’s milk	In vivo (mice models)	The low-density lipoprotein cholesterol levels, total serum cholesterol, liver lipids, coronary artery risk index, and liver TNF-α and IL-6 mRNA expression were reduced	[131]
*Lc. lactis* NRRL B-50571	Fermented cow’s milk	Clinical trial	The patient’s systolic and diastolic blood pressure and reduced blood serum triglyceride, total cholesterol, and low-density lipoprotein levels were reduced.	[132]
*L. acidophilus* La5 and *Bif. lactis* Bb12	Yogurt	Clinical trial	No effects were seen against heart disease	[133]
*L. acidophilus* La5 and *Bif. lactis* Bb12	Condensed yogurt (Kashk)	Clinical trial	Fat percentage, body mass index, and waist circumference were significantly reduced	[134]
*L. casei* Shirota	Yogurt	Clinical trial	The levels of glycol-albumins, low-density lipoproteins, and adipose tissue were decreased	[135]
*L. plantarum* nF1 heat-inactivated	Yogurt	In vivo (mice models)	Antiviral activity through activation of Natural Killer (NK) cytokine expression	[136]
*L. delbrueckii* ssp. *bulgaricus* OLL1073R-1	Fermented cow’s milk	In vivo (mice models)	Antiviral activity against influenza A virus	[137]

## Data Availability

Not applicable.

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
