# Peer review of "Health Benefits of Consuming Foods with Bacterial Probiotics, Postbiotics, and Their Metabolites: A Review"

_molecules, 2023, doi:10.3390/molecules28031230_

Round 1

Reviewer 1 Report

The manuscript is a review of Health benefits of consuming foods with bacterial probiotics,

postbiotics, and their metabolites.

About the corrections:

Figures: Authors should improve the quality, principally the font of the words.

The section 3.7 should be improved, due the importance of vitamins in probiotics

There is some type errors to correct.

Author Response

Reviewer 1

The manuscript reviews the health benefits of consuming foods with bacterial probiotics, postbiotics, and their metabolites.

Thank you for reviewing the manuscript and for your comments and suggestions to improve it.

About the corrections:

Figures: Authors should improve the quality, principally the font of the words.

R= The font in the figure (1, 2, and 3) was changed (enlarged), as well as the quality of the images.

Section 3.7 should be improved, due to the importance of vitamins in probiotics

R= More information was added about vitamins, such as physiological effects and relevant data for section 3.7.

There is some type errors to correct.

R= The type errors were identified and corrected.

Reviewer 2 Report

This manuscript was well-estblished, all tables and figures were in high quality and well-orgainzed, but I still have some commetns about this manuscript, as follows:

1.The application of bacterial probiotics, postbiotics, and their metabolites on food industry was less dsicussed.

2.The definition and compositions of probiotics, postbiotics, and their metabolites should be added in the revised manuscript.

3.Abstract and conclusion should be written and should be coordinated.

Author Response

This manuscript was well-established, and all tables and figures were of high quality and well-organized, but I still have some comments about this manuscript, as follows:

Thank you for reviewing the manuscript and for your comments and suggestions to improve it.

The application of bacterial probiotics, postbiotics, and their metabolites in the food industry was less discussed.

R= Section 5, “Probiotics and postbiotics in the food industry,” was included, which discusses the possible applications of probiotics and postbiotics.

The definition and compositions of probiotics, postbiotics, and their metabolites should be added in the revised manuscript.

R= Information about the definitions and compositions was added to the manuscript.

The abstract and conclusion should be written and should be coordinated.

R= The abstract was slightly rewritten to coordinate with the conclusion.